# Argonaute 2 targets viral transcripts but not genomes of RNA viruses during antiviral RNA interference in *Drosophila*

**Emanuele G. Silva**[1,2☉], **Isaque J. S. de Faria**[1☉], **Álvaro G. A. Ferreira**[1,3], **Thiago Henrique L. Jiran**[1,2], **Carlos F. Estevez-Castro**[1], **Juliana N. Armache**[1], **Siad C. G. Amadou**[1], **Yann Verdier**[4], **Joëlle Vinh**[4], **Karim Majzoub**[5], **Carine Meignin**[6], **Gabrielle Haas**[6], **Franck Martin**[7], **Jean-Luc Imler**[6], **João T. Marques**[1,2]*

1 Department of Biochemistry and Immunology, Instituto de Ciências Biológicas, Universidade Federal de Minas Gerais, Belo Horizonte, Minas Gerais, Brazil, 2 INSERM U1257, CNRS UPR9022, Université de Strasbourg, Strasbourg, France, 3 Mosquitos Vetores: Endossimbiontes e Interação Patógeno-Vetor, Instituto René Rachou-Fiocruz, Belo Horizonte, Minas Gerais, Brazil, 4 USR3149, ESPCI ParisTech, Paris, France, 5 Institut de Génétique Moléculaire de Montpellier, Université de Montpellier, CNRS UMR5535, Montpellier, France, 6 Université de Strasbourg, CNRS UPR9022, Strasbourg, France, 7 CNRS UPR9002, Université de Strasbourg, Strasbourg, France

☉ These authors contributed equally to this work.
* joao.marques@unistra.fr

## Abstract

RNA interference (RNAi) mediated by the small interfering RNA (siRNA) pathway is a major antiviral mechanism in insects. This pathway is triggered when double-stranded RNA (dsRNA) produced during virus replication is recognized by Dicer-2, leading to the formation of virus-derived siRNA duplexes. These siRNAs are loaded onto the programmable nuclease Argonaute-2 (AGO2), with one strand serving as a guide to target and cleave fully complementary sequences of viral RNAs. While siRNAs are generated from viral dsRNA, the specific viral RNA species targeted for silencing during RNA virus replication remains unclear. In this study, we characterized the primary viral RNA targets of the *Drosophila* siRNA pathway during infections caused by negative and positive RNA viruses, namely Vesicular stomatitis virus (VSV) and Sindbis virus (SINV). Our findings reveal that polyadenylated transcripts of VSV and SINV are the major targets of silencing by the siRNA pathway during infection, likely when they are poised for translation. Consistent with earlier findings, we show that AGO2 is associated with ribosomes in control and virus infected cells. Therefore, we propose that the inhibition of the replication of RNA viruses in *Drosophila* results from the silencing of incoming viral transcripts, facilitated by the association of AGO2 with ribosomes.

## Author summary

The small interfering RNA (siRNA) pathway mediates a major antiviral immune response in insects, functioning to cleave viral RNA. While this pathway has been extensively studied in the fruit fly *Drosophila melanogaster*, the specific molecular targets of inhibition by the siRNA pathway have remained unclear. In this study, we aimed to

**Data availability statement:** The authors confirm that all data underlying the findings are fully available without restriction. All relevant data are within the paper and its Supporting information files.

**Funding:** This work was supported by grants from Conselho Nacional de Desenvolvimento Científico e Tecnológico (CNPq) to J.T.M.; Fundação de Amparo a Pesquisa do Estado de Minas Gerais (FAPEMIG), Rede Mineira de Biomoléculas (grant # REDE-00125-16), Instituto Nacional de Ciência e Tecnologia de Vacinas (INCTV) to J.T.M.; Fonds régional de coopération pour la recherche FRCT2020 Région Grand-Est – ViroMod to J.T.M. and C. M.; Institute for Advanced Studies of the University of Strasbourg (USIAS fellowship 2019) to J.T.M.; and ANR-19-CE15-0021 to J-L.I. and J. T. M.. This study was financed in part by the Coordenação de Aperfeiçoamento de Pessoal de Nível Superior — Brasil (CAPES) — Finance Code 001 to J.T.M. This work of the Interdisciplinary Thematic Institute IMCBio, as part of the ITI 2021-2028 program of the University of Strasbourg, CNRS and Inserm, was supported by IdEx Unistra (ANR-10-IDEX-0002), by SFRI-STRAT'US project (ANR 20-SFRI-0012), and EUR IMCBio (IMCBio ANR-17-EURE-0023) under the framework of the French Investments for the Future Program to J.T.M, C.M. and J.-L.I.. E.G.S. was supported by fellowships from CNPq and CAPES. I.J.S.F. was supported by fellowships from CAPES and FAPEMIG. The funders had no role in study design, data collection and analysis, decision to publish, or preparation of the manuscript.

**Competing interests:** The authors have declared that no competing interests exist.

elucidate these targets. Our findings demonstrate that polyadenylated transcripts produced during viral infection are the primary targets of the *Drosophila* siRNA pathway, in the case of both negative and positive single-stranded RNA viruses. The silencing of these transcripts accounts for the antiviral effect of the siRNA pathway, suggesting that direct targeting of viral RNA genomes is unlikely to occur. We confirmed that Argonaute-2 (AGO2), the core component of the silencing complex, is in association with ribosomes regardless of viral infection. This suggests that AGO2 is in permanent association with ribosomes where it can efficiently scan viral transcripts before they undergo translation by the cellular machinery, thereby preventing viral replication. These results provide valuable insights into the mechanism of siRNA-mediated gene silencing.

## Introduction

The small interfering RNA (siRNA) pathway mediates a major antiviral defense in *Drosophila* and other insects [1]. This RNA interference (RNAi) pathway is triggered when the RNA helicase and RNase III enzyme Dicer-2 (Dcr-2) senses dsRNA generated during viral infection. Upon recognition, Dcr-2 processes dsRNA into siRNAs that are loaded onto Argonaute-2 (AGO2), a programmable nuclease, to form the RNA-induced silencing complex (RISC), which uses the sequence of the siRNA to find complementary targets [1,2]. The role of antiviral RNAi has been extensively characterized in the fruit fly *Drosophila melanogaster* during infection with a wide range of viruses with RNA or DNA genomes [3–9]. Infected flies accumulate virus-derived siRNAs (vsiRNAs) that, in the case of RNA viruses, cover both strands of the genome, indicating they are derived from dsRNAs intermediates of replication [7]. These vsiRNAs have the potential to target any viral RNA species in infected cells, both genomes and transcripts. Our previous work has suggested that the polyadenylated transcripts are preferentially targeted by RISC in infected *Drosophila* but this remains to be formally demonstrated [7]. Notably, silencing of either genomes or transcripts will affect virus replication and indirectly inhibit the production of all types of viral RNAs, which makes it hard to determine the initial target. In the case of *Tospovirus* infected plants, where RNAi is also a potent antiviral defense, viral transcripts were shown to be the primary target of silencing [10]. However, since the mechanism of antiviral RNAi is considerably different between plants and animals, it is unclear how generalizable this observation is. Here, we sought to identify the preferential targets of the siRNA pathway during RNA virus infection in *Drosophila*.

## Results

Vesicular Stomatitis virus (VSV) is a negative-stranded RNA virus of the *Rhabdoviridae* family that induces a potent siRNA response in *Drosophila* [7,9]. In order to evaluate the specificity and persistence of antiviral responses, we developed an *in vivo* strategy where flies were primed with replication competent or UV-inactivated VSV before being exposed to a secondary infection with the same virus (Fig 1A). To differentiate primary and secondary infections, we used a recombinant virus expressing GFP (VSV-GFP) on the latter. Expression of GFP was used as a reporter for replication of the second virus while the L gene encoding the VSV polymerase reflected replication of both primary and secondary infections. We observed that the replication of VSV-GFP was strongly inhibited by prior exposure to the replication competent VSV but not UV-inactivated virus (Fig 1B). We reasoned that dsRNA intermediates generated only by replication competent viruses would trigger the siRNA pathway to restrict the replication of the second homologous virus. To address this hypothesis, we synthesized

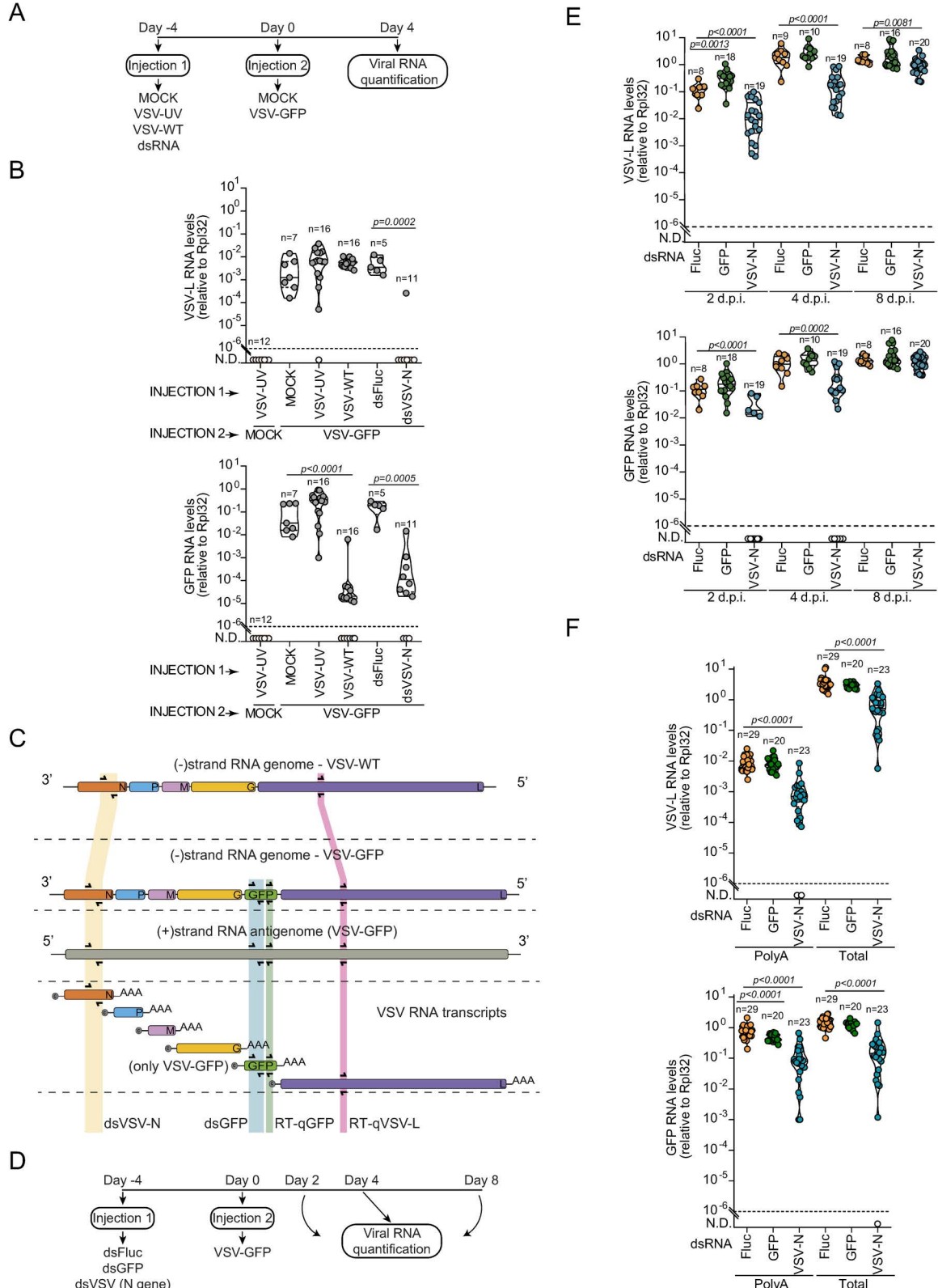

**Fig 1. Polyadenylated viral transcripts are preferential targets for dsRNA mediated silencing during VSV infection. (A)** Experimental design of double infections and dsRNA treatment followed by VSV infection in *Drosophila*. **(B)** VSV-L and GFP RNA levels at 4 days post infection with VSV-GFP. **(C)** Overview of VSV-WT and VSV-GFP RNA species and the target regions selected for dsRNA

synthesis (dsVSV-N and dsGFP) and RT-qPCR assays (qGFP and qVSV-L). **(D)** Experimental design of dsRNA treatment followed by VSV infection in *Drosophila*. **(E)** VSV-L and GFP RNA levels at the indicated times post infection with VSV-GFP. **(F)** Levels of polyadenylated and total RNAs corresponding to VSV-L and GFP genes at day 4 post infection with VSV-GFP. RNA levels were assessed by RT-qPCR performed in triplicate and normalized to Rpl32 mRNA levels. Violin plots show the frequency of data distribution with the median and quartiles. Each dot represents an individual sample. Numbers of samples are indicated above each violin plot. The dashed line represents the detection threshold of RT-qPCR. Significant *p* values *(p < 0.05)* determined by two-tailed Mann–Whitney U-test are shown.

dsRNA targeting the nucleoprotein gene (N) from VSV (dsVSV-N) and compared to dsRNA targeting an unrelated gene, *Firefly luciferase*, as a control (dsFluc). Injection of dsVSV-N into *Drosophila* adults induced ~1,000-fold decrease in the levels of the VSV L gene and GFP RNAs compared to dsFluc-treated flies (Fig 1B). Taken together, our results suggest that the protection of flies triggered by replication competent VSV is likely mediated by the detection of dsRNAs by Dcr-2, which result in the production of virus-specific siRNAs that control infection by homologous viruses.

Since synthetic dsRNA targeting VSV primes antiviral RNAi, we decided to use our model to further understand how the siRNA pathway targets viral replication. Capped and polyadenylated transcripts as well as genomes and antigenomes are produced during VSV infection by the viral RNA dependent RNA polymerase (RdRp) (Shown in Fig 1C) [11–13]. In our model, GFP reflects viral replication, but production of the protein is not essential for replication. Thus, if polyadenylated transcripts encoding GFP are targeted by the silencing machinery, this would not affect VSV replication. In contrast, if viral genomes or antigenomes were to be targeted, silencing of the GFP within the viral genomic RNA would block replication. Thus, we injected dsRNA targeting GFP (dsGFP), dsVSV-N and dsFluc into distinct groups of flies before VSV infection and collected them at different time points to access viral loads (Fig 1D). dsVSV-N strongly inhibited VSV replication, as shown by decreased expression of the VSV-L gene and GFP throughout the kinetics of infection (Fig 1E). In contrast, treatment with dsGFP did not decrease levels of VSV L (Fig 1E). These results suggest that siRNAs generated from dsRNA processing did not target the VSV-GFP genome since it did not decrease virus replication. Notably, dsGFP did not significantly decrease levels of the GFP RNA in VSV-GFP infected cells (Fig 1E), which we attribute to the fact that GFP levels are a combination of transcripts and RNA genomes. Thus, we next measured the abundance of polyadenylated RNAs to directly evaluate levels of viral transcripts. Similar to the results measuring total RNA, levels of polyadenylated RNAs corresponding to L gene and GFP were significantly reduced by treatment with dsVSV-N (Fig 1F). In contrast, dsGFP treatment did not affect levels of VSV-L but reduced levels of the GFP transcript produced from the recombinant virus (Fig 1F). dsGFP also efficiently silenced GFP expression in adult flies where the transgene is controlled by a hemocyte specific promoter, showing that it efficiently triggers silencing (S1 Fig). Taken together, these results indicate that viral transcripts and not genomic RNAs are the major target of the siRNA pathway during VSV infection in *Drosophila*.

To investigate whether our observations could be applied to other viruses, we used a positive stranded RNA virus, Sindbis virus (SINV), that belongs to the *Togaviridae* family. The single-stranded RNA genome of SINV also serves as an mRNA for the translation of nonstructural proteins (nsPs) while a subgenomic RNA encodes the structural proteins (sP) [14,15]. In our experiments, we used a version of SINV that expresses GFP (SINV-GFP) as an independent subgenomic RNA that is dispensable for viral replication. We analyzed the kinetics of SINV-GFP replication in flies injected with dsRNA targeting nsP4, encoding the RdRp of SINV (dsSINV-nsP4), compared to injection of dsGFP and the unrelated dsFluc as a control (Fig 2A–2C). We observed a 10 to 100-fold reduction in the viral load of SINV

as reported by the levels of another SINV gene, nsP2, at all times post-infection. We also observed a reduction of the levels of SINV sP and GFP in flies treated with dsSINV-nsP4 after 2 days post-infection, indicative of general inhibition of viral replication (Fig 2C and 2D). In contrast, dsGFP did not consistently affect the viral load of SINV as indicated by the levels of nsP2 or sP, but it significantly decreased the subgenomic RNA encoding GFP (Fig 2C). We observed a significant decrease in nsP2 levels in flies treated with dsGFP at 8 days post infection (Fig 2C). However, dsGFP did not have a strong effect compared to direct silencing of nsP4, an essential viral gene (Fig 2C). Thus, targeting GFP mostly affected its own expression levels but not replication of SINV-GFP. Considering that the positive strand of the SINV genome can be directly translated as a transcript, it was unexpected to have such a small effect on viral replication when GFP was targeted. This indicates that dsGFP mostly triggered silencing of its own subgenomic RNA, leaving unaffected the full-length genomic RNA, suggesting that the silencing machinery may only be able to scan targets close to the ORF that is being translated.

Previous studies in *Drosophila* S2 cells found that AGO2, the main component of RISC, is associated with cellular ribosomes [16–18]. To confirm these observations during viral infection, we separated ribosome enriched (P100) and depleted (S100) fractions of S2 cells (Fig

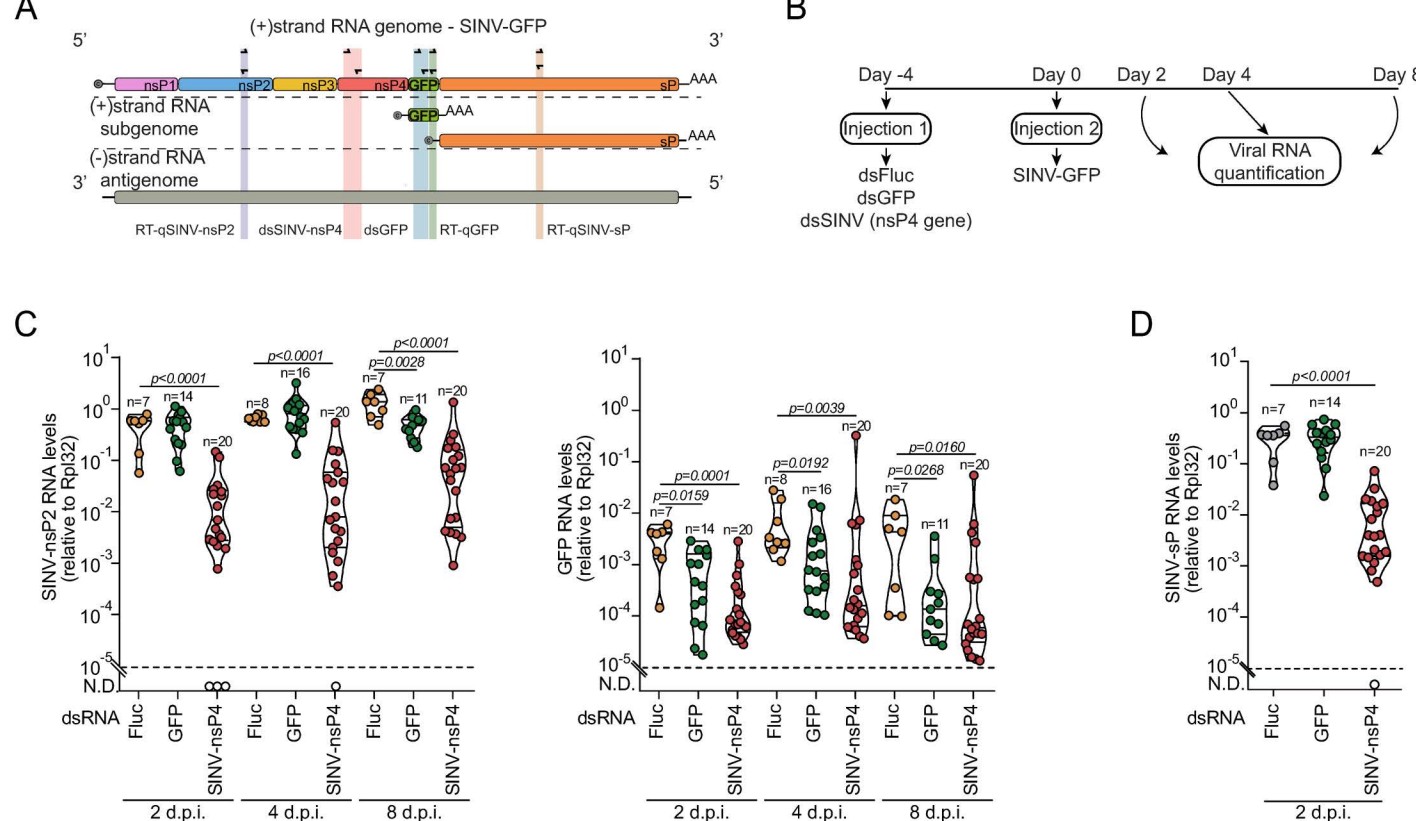

**Fig 2. Translated RNAs are preferential targets for dsRNA mediated silencing during SINV infection.** **(A)** Overview of SINV-GFP RNA species and the target regions selected for dsRNA synthesis (dsSINV-nsP4 and dsGFP) and RT-qPCR assays (qSINV-nsP2, qSINV-sP and qGFP). **(B)** Experimental design of dsRNA treatment followed by SINV-GFP infections in *Drosophila*. **(C)** SINV-nsP2 and GFP RNA levels at the indicated times post infection with SINV-GFP. **(D)** SINV-sP RNA levels at day 2 post infection with SINV-GFP. RNA levels were assessed by RT-qPCR and normalized to Rpl32 mRNA levels. Violin plots show the frequency of data distribution with the median and quartiles. Each dot represents an individual sample. Numbers of samples are indicated above each violin plot. The dashed line represents the detection threshold for the RT-qPCR. Significant *p* values *(p<0.05)* determined by two-tailed Mann–Whitney U-test are shown.

3A), as reported before [19]. In both VSV infected and control cells, the ribosomal protein S15 (RpS15) was enriched in the P100 fraction, confirming that ribosomes were efficiently separated by our protocol (Fig 3B). Regardless of the infection, AGO2 was present in the P100 sample (lines 2 and 4) and depleted from S100 fractions (lines 1 and 3), confirming that it is in the same fraction as ribosomes, regardless of the infection (Fig 3B). Interestingly, in VSV infected cells, the viral RdRp was found both in ribosomal enriched and depleted fractions, possibly indicating multiple complexes responsible to the synthesis of the different viral RNA species (Fig 3B). Using confocal microscopy, we observed a significant majority of AGO2 in the cytoplasm while ~20% was nuclear (Fig 3C and 3D), as reported before [20]. In the cytoplasmic fraction, we observed that ~70% of AGO2 overlapped with RpS15 while the rest was present in zones without ribosomes (Figs 3D and S2A). In accordance with a direct association with ribosomes, we observed that AGO2 interacted with many ribosomal proteins based on immunoprecipitation assays (Fig 3E), as previously described [18]. Enrichment analysis showed a significant enrichment of ribosomes and other processes in AGO2 immunoprecipitates in both infected and non-infected cells, although infection seems to decrease the significance of the association (Fig 3F). Ribosomal proteins interacting with AGO2 were mostly localized around the mRNA channel with some differences between infected and control cells (S2B Fig). There were also other changes in AGO2 associated proteins between infected and control cells, possibly pointing to a change in function during infection (Fig 3F).

## Discussion

Here, our results suggest that AGO2 is closely associated with ribosomes where it can easily monitor incoming target mRNAs to promote their degradation. This likely determines the preferential targeting of viral transcripts rather than genomes by antiviral RNAi (Fig 3G). Similarly, previous work in *Trypanosoma brucei* has shown that a fraction of its argonaute is found in association with polysomes where it can more efficiently silence target mRNAs [21,22]. Our new observations are consistent with results suggesting that polyadenylated viral RNAs from VSV and SINV are preferentially silenced compared to genomic RNAs [7]. Similarly, RISC-associated nuclease activity was shown to act at the level of viral transcripts but not genomic RNA in *Tospovirus* infected plants [10]. In this case, authors suggested that viral transcripts are preferentially targeted by the RNAi machinery because they are not protected within ribonucleoprotein complexes unlike viral genomes/antigenomes. Our results also make sense considering that unmasking target sites by translating ribosomes seems to be the limiting step in human AGO2 mediated mRNA degradation [23]. This may be even more significant in the case of *Drosophila* AGO2 that has a faster turnover rate than human AGO2 [24,25]. Thus, for *Drosophila* AGO2 that can catalyze multiple rounds of RNA cleavage, availability of the target is certainly the limiting factor in silencing efficiency. We propose that the RISC complex is constantly present at translation sites, where it could most efficiently scan incoming viral mRNAs at the step where they are most exposed and before they are translated into viral proteins. These results provide new insights on the control of viral infection by RNAi in animals, which points to the dependency on ribosomes as a central weakness for viruses.

## Materials and methods

### Virus stocks and cells

VSV-WT was a gift from Dr. Erna Geessien Kroon (strain Indiana isolate P94). VSV-GFP and SINV-GFP were a gift from Curt Horvath and Ilya Frolov, respectively. For virus propagation,

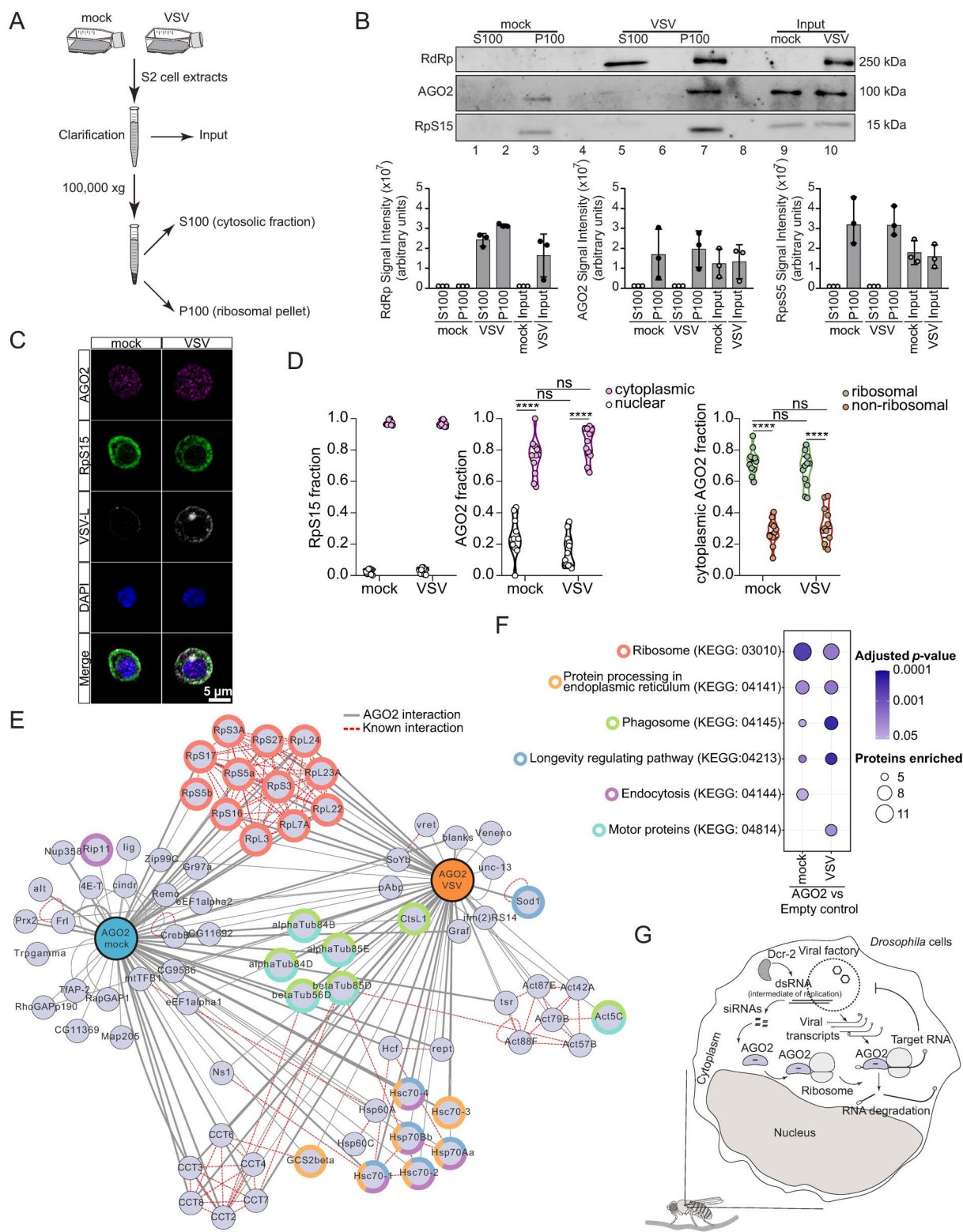

**Fig 3. AGO2 is associated with ribosomes in control and virus infected cells.** (A) Overview of the cell fractionation strategy. Mock- or VSV-infected cell lysates were submitted to high-speed centrifugation (100,000 g). The S100 and P100 containing the cytoplasmic and ribosome fractions, respectively, were extracted with high salt buffer and proteins were analyzed by Western blotting. (B) Representative Western blotting

(upper panel) and quantification of signal intensity (lower panel) for VSV-RdRp, AGO2 and RpS15 proteins of S100 and P100 fractions and input. Graphs represent the means ± SD of three independents experiments indicated by the dots. **(C)** Immunofluorescence analysis showing sub-cellular localization of AGO2, RpS15 and the VSV polymerase (RdRp) encoded by the L gene in control and infected S2 cells. Scale bar = 5 μm. **(D)** Quantification of nuclear and cytoplasmic localization of RpS15 and AGO2 in the left and middle panels, respectively. The right panel shows the fraction of cytoplasmic AGO2 overlapping with RpS15 (ribosomal) or not (non-ribosomal). **(E)** Network analysis of AGO2 interacting proteins in infected and non-infected cells based on immunoprecipitation followed by mass spectrometry. Biological pathways associated with interactants are indicated by color as in panel F. **(F)** Enrichment of biological pathways in AGO2 immunoprecipitates from infected and non-infected cells. *p*-value of the enrichment and the number of proteins representing each enriched biological pathway are indicated. **(G)** Overview of the molecular mechanism of inhibition by the siRNA pathway, which preferably targets viral RNA transcripts during infection with RNA viruses in *Drosophila*.

VERO cells were maintained on DMEM medium (LGC) supplemented with 10% heat-inactivated Fetal Bovine Serum (FBS, Gibco), 1XGlutaMAX (Gibco) and 1X penicillin/streptomycin (10 mg.mL 1/10,000 U, Gibco). Cells were seed to 80% of confluence and infected with VSV-WT or VSV-GFP at multiplicity of infection of 0.001 and for SINV-GFP, at 0.1. Cells were maintained for 20–36 h at 37ºC until the appearance of cytopathic effects. After this time, the supernatant was collected and clarified by centrifugation to generate virus stocks that were kept at −80 °C before use. Mock-infected supernatants were prepared under the same condition without virus infection as control of experiments. Titration by plaque assay of VSV-WT and VSV-GFP was performed on VERO cells and for SINV-GFP on BHK-21 cells. VSV-WT inactivation was done by UV radiation at the given 254 nm wavelength as previously described [26]. VSV-UV was titrated after UV exposure to verify viral inactivation. *Drosophila* S2 cells were maintained in Schneider's medium supplemented with 10% (FBS, Gibco), 1XGlutaMAX (Gibco), and 1X penicillin/streptomycin (10 mg.mL 1/10,000 U,Gibco). Cells were incubated at 28ºC for viral infections.

### *Drosophila* and S2 infections

Fly stocks were raised on standard cornmeal-agar medium at 25 °C. Adult females flies 3 to 6 days of age were used in infection experiments *in vivo*. *Canton S* genotype flies were obtained from the Bloomington Fly Stock Center (Bloomington, IN). Infections were done by intrathoracic injection (Nanoject II apparatus; Drummond Scientific) of a 69 nL of 150 ng of dsRNAs or viral suspension (5,000 PFU/fly for VSV-WT and 500 PFU/fly for VSV-GFP and SINV-GFP). Injected flies with dsRNAs or viruses were maintained at 25ºC for different times according to each experiment setup. After these times, flies were harvested in TRIzol reagent (Invitrogen). Samples were homogenized using a mini-Bead beater homogenizer (BioSpec). For *in vitro* experiments, $10^7$ S2 cells were counted, transferred to canonical tubes and infected with VSV-GFP at MOI of 1, at 28ºC under gentle homogenization. After 1 h, cells were spun down at 200 g, at 4ºC for 5 min and the medium removed. Cells were resuspended in supplemented Schneider's medium and seeded into T25 culture dishes. Infected cells were incubated at 28ºC for 24 h and submitted to cell fractionation.

### Quantification of viral RNAs by RT-qPCR

Total RNA from adult flies or S2 cells were extracted using Trizol reagent according to the manufacturer's protocol (Invitrogen). 500 ng of total RNA were reverse transcribed using 300 ng of random primers or 500 ng of anchored oligo dT primers. 2 uL of diluted cDNA were used as template for qPCR reaction containing Sybr Green (Invitrogen) and specific primers. RT-qPCR was performed using QuantStudio 7 Flex RealTime PCR System (Applied Biosystems). Analyses of RNA expression was done using $\Delta C_T$ method with Rpl32 as the internal control gene. Primer sequences are provided in S1 Table.

## dsRNA constructions and treatments

*In vitro* transcription was done using T7/SP6 MEGAscript Kits (Ambion), following the manufacturer's instructions. Briefly, dsRNAs targeting viral sequences were produced from a T7 promoter sequences obtained by PCR amplification from TOPO plasmid (Thermo #K4575J10) containing purified PCR product corresponding to N region of VSV gene (region: 427–962) or to nsP4 region of SINV gene (region: 5,920–6,424). dsFluc and dsGFP were produced from a T7 and SP6 promoters sequences containing the firefly luciferase sequence from pGL3-Basic plasmid (Promega) and GFP PCR amplification from plasmid pDSAG (Addgene #62289) respectively. Adult female flies were intrathoracically injected with 69 nL of a dsRNA solution (150 ng/fly) diluted in annealing buffer (20 mM Tris-HCl pH7.5, 100 mM NaCl). Primers sequences for dsRNAs construction are provided in S1 Table.

## Cell fractionation

Cell fractionation was performed as described [19]. Briefly, log-phase S2 cells were plated on T25 culture dishes and were VSV or mock- infected. After 24 h, cells were harvested in PBS containing 5 mM EGTA and washed twice in cold PBS and once in hypotonic buffer (10 mM HEPES pH 7.3, 6 mM b-mercaptoethanol). Cells were suspended in 1 mL of hypotonic buffer containing complete protease inhibitors (Protease Inhibitor Cocktail Tablets, EDTA-Free, Roche) and 0.5 units mL−1 of RNAseOUT (Invitrogen) and then disrupted in a tissue homogenizer with a type B pestle. The extract was centrifuged at 800 g for 5 min at 4ºC for cell clarification and the supernatant was subsequently centrifuged at 100,000 g for 3 h. The resulting pellet, containing ribosomes, was extracted in hypotonic buffer containing 1 mM $MgCl_2$ and 400 mM KOAc. The cytosolic fraction (S100) and ribosome fraction (P100) were used to quantify proteins.

## Western blotting

After cell fractionation, S100, P100 and input were resuspended in 6X Laemmli SDS PAGE sample loading buffer. Total cellular protein for each sample was subjected to SDS-PAGE, followed by electroblotting onto nitrocellulose membranes. Membranes were blocked with 5% w/v BSA or 10% w/v nonfat dry milk in TBST buffer (150 mM NaCl, 10 mM Tris–HCl, pH 7.4, and 0.1% Tween 20) for 1 h and then incubated with anti-rabbit RpS15 (Abcam#168361), anti-rabbit AGO2 (Abcam#32381) or guinea-pig anti-VSV (raised against the recombinant protein comprising the 188 N-terminal amino acids of VSV RdRp by Protéogenix) antibodies in TBST buffer containing 3% w/v BSA or 5% w/v nonfat dry milk overnight at 4°C. Membranes were rinsed three times with TBST buffer and incubated with the related secondary peroxidase conjugated anti-IgG antibody diluted 1:5,000 in TBST buffer containing 3% w/v BSA or 5% w/v nonfat dry milk for 1 h. Membranes were rinsed three times with wash buffer, incubated with ECL prime western blotting detection reagents, and scanned and analyzed by ImageQuant LAS 4000 (GE Healthcare).

## Immunofluorescence

*Drosophila* S2 cells were mock infected or infected with 1 PFU/cell of VSV for 72 hours then transferred to coverslips treated with Poly-D-lysine (Gibco, Cat# A3890401) and incubated for 30–60 min to allow cell attachment. Cells were then fixed with 4% formaldehyde (Pierce) for 10 min, washed three times in PBS and permeabilized with 0.1% Triton X-100 in PBS (PBST) for 20 min. Cells were blocked with 10% of FBS (Gibco, Cat# 12676029) in PBST (blocking buffer) for 1 h at 25ºC. Primary antibodies include mouse anti-AGO2 (1/10 dilution of anti-AGO2 supernatant, a gift from M. Siomi and H. Siomi), guineapig anti-VSV and

rabbit anti-RpS15 (1/200 dilution, Abcam ab157193), all used in blocking buffer for overnight incubation with cells at 4°C. Cells were then washed three times with PBST. Secondary antibodies include Goat anti-Guinea Pig IgG (H+L) Highly Cross-Adsorbed Secondary Antibody, Alexa Fluor 488 (Cat# A-11073, RRID:AB_2534117), Donkey anti-Rabbit IgG (H+L) Highly Cross-Adsorbed Secondary Antibody, Alexa Fluor Plus 555 (Cat# A32794, RRID:AB_2762834), Donkey anti-Mouse IgG (H+L) Highly Cross-Adsorbed Secondary Antibody, Alexa Fluor Plus 647 (Cat# A32787, RRID:AB_2762830). All secondary antibodies were used at 300-fold dilution in PBST associated to DAPI DNA staining (Cat# D3571, Molecular Probes) and incubated with cells for 1 h at 25°C in dark. Coverslips were washed three times with PBST and rinsed once with PBS. Coverslips were mounted in microscope slides with Hydromount medium (National Diagnostics) and sealed with nail polish. Cells were visualized using Airyscan mode of a Zeiss LSM880 confocal microscope (Carl Zeiss Microimaging, Inc.) using an alpha Plan-Apochromat 100X/1.46 Oil DIC M27 Elyra objective. Z stacks were acquired with Z-slices spaced at 0.17 μm intervals. AGO2, RpS15 and VSV-L staining were crosschecked for non-specific staining using controls with secondary but not primary antibody incubations as channel controls. Image stacks containing 1–2 cells/image were acquired with 4 separate channels run in the following order, AF647, AF555, AF488, and DAPI. Raw Airyscan images were processed with the Airyscan processing application in Zen software (Zeiss), leaving the Processing strength into "auto" mode, using the 3D option. Processed images were saved as CZI and analyzed in Image J. For 3D-colocalization analysis in ImageJ, the channels were separated, converted into 8-bit images, and AGO2 and RpS15 channels subject to Background subtraction using a Rolling ball radius of 50 pixels, with Sliding paraboloid mode on, running throughout all slices in the stack. AGO2, RpS15 and DAPI channels were merged into a single multi-channel image and the overlapping pixels (colocalization) were analyzed using BIO JACoP plugin (Automatic threshold Otsu method, all Z slices considered, Costes block size (pixel) = 5, Costes Number of Shuffling = 100, Colocalization Result Options: Get Manders Coefficients). Thresholder M1 and M2 from Manders Coefficients representing AGO2 and ribosomes (RpS15) fractions were then selected for Statistical analysis and graphical representation in GraphPad Prism (GraphPad, La Jolla, CA, USA) using 2-way ANOVA with Sidak's multiple comparisons test. Nuclear fractions were calculated by the percentage of AGO2 and RpS15 staining colocalizing with DAPI, and cytoplasmic fractions are represented by 1 (total) - (nuclear fraction). Cytoplasmic AGO2 fractions in non-ribosomes correspond to 1 (total) - (RpS15-AGO2 fractions). All the raw data is shown in S2 Table. For visual representation, the sum of Z slices was processed for individual channels, AGO2, RpS15 and VSV-L channels subject to background subtraction and representative images from mock and VSV-infected cells are displayed.

## Protein purification, identification and quantification by mass spectrometry

cDNA from AGO2 gene was amplified by standard PCR and inserted into a pDONR221 vector (Invitrogen) using Gateway cloning technology. PCR fragments were recombined (BP reaction) with pDONR221 to obtain pENTRY-tag-AGO2 (we used two individual tags on the N- or C-terminus). We then generated an OpIE2 insect expression vectors based on the pIZ/V5-His backbone containing the biotin-tag sequence (encoding the GLNDIFEAQKIEWHE peptide). BirA-expressing *Drosophila* S2 cells expressing biotinylated-AGO2 protein were either mock infected or infected with VSV at MOI 10 for 48 h. Protein purification and identification was performed as described [27]. Briefly, 20 million cells for each condition were lysed in 1 mL of TNT buffer (50 mM Tris-HCl, pH 7.5, 150 mM NaCl, 10% Glycerol, 1% Triton X-100, 100 mM NaF, 5 μM $ZnCl_2$, 1 mM $Na_3VO_4$, 10 mM EGTA, pH 8.0, Complete

Protease Inhibitor Cocktail containing EDTA from Roche). Lysates were kept on ice for 30 min then centrifuged at 16 000g for 30 min at 4°C. Supernatants were mixed with 150 µl of prewashed streptavidin–sepharose beads and incubated for 30 min at 4 °C in a rotative agitator. Beads were washed three times with 1 mL Wash buffer I (50 mM Tris–HCl, pH 7.5, 150 mM NaCl, 10% Glycerol, 0.1% Triton X-100, 100 mM NaF, 5 µM $ZnCl_2$, 1 mM $Na_3VO_4$, 10 mM EGTA, pH 8.0), one time with 1 mL Wash buffer II (Wash buffer I without Triton X-100), and suspended in 1 mL Wash buffer II plus Complete Protease Inhibitor Cocktail containing EDTA. The on-beads tryptic digestion method was used on the purified samples. After reduction/alkylation (DTT, 5 mM final, 30 min, 56°C/ Iodoacetamide 25 mM final, 20 min in the dark), 10 ng trypsin (modified sequencing grade, Roche) in 150 mM ammonium carbonate was added and samples were incubated overnight at 37°C with shaking. Then, the reaction was stopped with 10 µL 10% formic acid (FA). Peptides were recovered and the beads were discarded by filtering through C18 Tips (Proxeon). Peptides were eluted with 20 µL 50% methanol, 5% FA. 5 µL of peptides diluted 1:5 in water (5% of the total amount of material) were purified on a capillary reversed phase column (nano C18 Acclaim PepMap100 Å; 75 µm internal diameter, 15 cm length; Dionex), at a constant flow rate of 220 nL.min$^{-1}$, with a gradient 2% to 40% buffer B (water/acetonitrile/FA 10:90:0.1 (v:v:v)) in buffer A (water/ aceto-nitrile/FA 98:2:0.1 (v:v:v)) over 45 min. The first MS analysis was performed on a FT ICR mass spectrometer (LTQ-FT Ultra, ThermoFisher Scientific, San Jose, CA) with the top 7 acquisition method: MS resolution 60,000; mass range 500–2,000 Th; followed by 7 MS/MS (LTQ) on the 7 most intense peaks, with a dynamic exclusion for 90s. The raw data was processed using Xcalibur 2.0.7 software and Mascot daemon. Each sample was first analyzed in triplicate then an exclusion list was added for three next runs. The database search was done on merged data using Mascot search engine (Matrix Science Mascot 2.3) on the 17 *D. melanogaster* database (16,535 sequences) concatenated with protein sequences of VSV. Proteome Discoverer 1.3 (ThermoFisher Scientific) and Mascot were used to search the data and filter the results. The following parameters were used: up to 2 miss cleavages; MS tolerance 10 ppm; MSMS tolerance 1Da; full tryptic peptides; partial modifications: carbamidomethylation (C), oxidation (M, H, W), Phosphorylation (Y). Validation was performed on proteins identified using two filters: i) only proteins identified with an FDR < 1% (Peptide Validator Mascot significance threshold) and at least 1 peptide score above 30 were selected; ii) only proteins identified with 2 distinct sequences with ion score above 30, or 1 sequence with ion score above 30 with and a MudPIT (Multidimensional Protein Identification Technology score) score above 49. Proteins identified by a peptide matching another protein were not considered and were filtered out for gene selection and subsequent validation tests. Proteins that passed filter ii) and not filter i) were manually checked: they were validated if the identified sequence was specific to the associated gene after control of the MS/MS spectrum. Protein that passed filter i) and not the more stringent filter ii) were not considered. Full results are shown in S3 Table.

## Analysis of mass spectrometry data

MS/MS data was analyzed using a presence/absence approach, including all validated hits from both tags (AGO2 N- or C-terminus) in each condition (mock and VSV) that were not detected in the respective control IP (Empty vector, mock and VSV). The integrated interactome network was visualized using Cytoscape v3.10.2 and Cytoscape StringApp [28], integrating experimentally determined protein-protein interactions retrieved from the Molecular Interaction Search Tool (MIST) database [29]. Edge width was determined by the number of hits between the paired experiments (N- or C-terminus). Gene Ontology (GO) enrichment analysis was conducted using g:Profiler [30], applying the g:SCS algorithm with an adjusted *p*-value threshold of 0.05. Enrichment results were plotted as a bubble plot using ggplot2 v3.5.0 in R (v4.3.0, R Core

Team, 2021). Structural visualization of ribosomal interactants of AGO2 was performed using the *Drosophila melanogaster* Ovary 80S ribosome structure (PDB ID: 6XU8) [31] in PyMOL (v2.4.0, The PyMOL Molecular Graphics System, Schrödinger, LLC).

### RT-qPCR analysis

GraphPad Prism software was used to analyze data for statistical significance in viral RNA loads compared to control groups. A two-tailed student t test was used to statistically analyze the data determined by unpaired *t*-test followed by Mann-Whitney post-test. All raw data for qPCR analysis is shown in S2 Table.

## Supporting information

**S1 Fig. Silencing of GFP expressed in transgenic flies.** Experimental design of dsGFP treatment in *Hml-delta-GAL4, UAS-GFP* flies in the left panel. Right panel shows GFP RNA levels at day 3 post treatment with dsGFP. Significant *p* value determined by two-tailed Mann–Whitney U-test is shown.
(PDF)

**S2 Fig. AGO2 is mostly located in the cell cytoplasm in association with ribosomes. (A)** Immunofluorescence showing staining for AGO2, RpS15 and the VSV polymerase (RdRp) in infected and control S2 cells. Scale bar = 20 μm. **(B)** Structure of the *Drosophila* ribosome highlighting the subunits found to interact with AGO2 in control and VSV-infected cells (common interactants). Ribosomal subunits interacting with AGO2 in only one condition, mock or VSV-infected cells, are shown in different colors. The mRNA channel is indicated.
(PDF)

**S1 Table. Primers sequences utilized for RT-qPCR assays and dsRNA synthesis.**
(PDF)

**S2 Table. Minimal dataset for submission.**
(XLSX)

**S3 Table. AGO2 interactants identified by mass spectrometry.**
(XLSX)

## Acknowledgments

The authors would like to thank members of Marques and Imler groups for discussion and suggestions.

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
