## [Decision Letter · Decision Letter 0]

16 Jul 2024

Dear Dr. Marques,

Thank you very much for submitting your manuscript "Antiviral RNA interference targets viral transcripts but not genomes of RNA viruses in Drosophila melanogaster" for consideration at PLOS Pathogens. As with all papers reviewed by the journal, your manuscript was reviewed by members of the editorial board and by several independent reviewers. I would mention that it was extraordinarily difficult to get reviewers to undertake this assignment, hence the delay.  In light of the reviews (below this email), we would like to invite the resubmission of a significantly-revised version that takes into account the reviewers' comments.

Since both reviewers mentioned the need for more confirmative data regarding the association go Ago2 with ribosomes, I suggest that something be done to address this particular comment. 

We cannot make any decision about publication until we have seen the revised manuscript and your response to the reviewers' comments. Your revised manuscript is also likely to be sent to reviewers for further evaluation.

Sincerely,

Benhur Lee

Section Editor

PLOS Pathogens

Benhur Lee

Section Editor

PLOS Pathogens

Michael Malim

Editor-in-Chief

PLOS Pathogens

orcid.org/0000-0002-7699-2064

Reviewer's Responses to Questions

**Part I - Summary**

Reviewer #1: In this study, Silva et al. characterized the primary viral RNA targets of the Drosophila siRNA pathway during infections caused by VSV and SINV. Their data demonstrated that polyadenylated transcripts of VSV and SINV are the primary targets of silencing by the siRNA pathway during infection. They also stated that AGO2 copurifies with ribosomes, a process unaffected by virus infection. These findings enhance our understanding of the antiviral mechanism executed by RNAi silencing in Drosophila, particularly the specific viral RNA species targeted during RNA virus infection. Overall, the manuscript is well-written. However, the major findings are based on incomplete solid evidence. Additional experiments are needed to support the manuscript's major claims.

Reviewer #2: This manuscript presents a series of experiments demonstrating that Drosophila Ago2-mediated antiviral RNAi (primarily) targets the polyA ‘mRNA’-like transcripts, rather than genomes, of VSV (a -ssRNA virus) and SINV (a +ssRNA virus). The authors further argue that the strong association of Ago2 with a ribosomal fraction suggests a scanning mechanism whereby Ago2 interacts with transcripts as they are read by the ribosome.

The manuscript is well written, and I think the work is sound as far as it goes: the experiment is simple, but clever, and I could not immediately identify any faults in the approach or logic. My only concerns with scale and importance of the finding (is this big enough for PLoS Pathogens, even as a short report). Namely, the targeting of transcripts was already suspected, and previously shown in plants; a ribosomal association has been previously reported, but an artefactual association is not strongly ruled out here; the proposed scanning mechanism based on ribosomal-association is nice, but not really tested and remains hypothetical.

This is clearly an editorial decision, but on balance I think the paper is indeed suitable. Drosophila is a (the?) major model for antiviral RNAi in animals, the targeting of mRNAs is shown here quite compellingly, and the mechanistic hypothesis is a good basis for further research. I have no doubt that this work will be read and cited by those working on antiviral RNAi and RNAi mechanisms more generally.

**Part II – Major Issues: Key Experiments Required for Acceptance**

Reviewer #1: 1. Figure 1B: The authors injected VSV-WT at 5000 PFU/fly for the first experiment and VSV-GFP at 500 PFU/fly for the second. Intuitively, these viruses should replicate at higher levels given the 10X difference in viral load. However, the VSV-L RNA levels in these injected flies were similar to those in flies first injected with Mock, VSV-UV, and dsFluc. This result is difficult to understand.

2. Figures 3A and 3B: The cell fractionation strategy described is too simplistic to provide detailed insights, even in the Methods section. According to the manuscript, the final P100 pellet contains not only ribosomes but also other organelles and protein complexes. In Figure 3B, the authors used Rps15 as a ribosome marker. However, this result only proves that ribosomes are present in the P100 pellet; it does not confirm that AGO2 is associated with ribosomes. Additional experiments are needed to demonstrate AGO2's association with ribosomes.

Reviewer #2: None

**Part III – Minor Issues: Editorial and Data Presentation Modifications**

Reviewer #1: Figure 1C: Please verify the positions of the dsRNA and qPCR primers.

Reviewer #2: My specific (minor) comments are listed below:

1. The title seems a little odd, as I would consider the dicing action of dicer2 to be a part of RNAi, and that does target the genome (during replication)

2. Line 41: ‘a’ or ‘the’ major?

3. Line 48: this feels like an overclaim. Is there not some impact of Dicer2 activity on virus replication or copy number, regardless of the role of Ago2? I think I recall an Ago2 K/O paper that claimed Ago2 but not Dcr2 was dispensable for successful response to DCV (perhaps because of the VSRs encoded by DCV?). Although now I can’t find which paper …

4. L67. Do the authors think that their findings might bear on the patchy (along the genome) and biased (between the strands) distribution of viRNAs seen in many viruses? If so, it could eb added to the discussion and siRNA properties introduced here.

5. L123-125. The small effect size suggests a very small proportion of the RNA is transcript rather than genome (am I correct in this interpretation?) – is that credible, from what is known of the replication of this virus?

6. L152-155 This observation, of the apparently highly localised nature of the scanning, is interesting and warrants more discussion and more emphasis. If I were to suggest expanding the paper with further experiments, this would be a good place to do it.

7. L165-167 and elsewhere. The association with ribosomes is interesting, but for a high-copy entity like a ribosome, it is always a concern that this association may be spurious. It would be very helpful to know what other (non-ribosomal) proteins are enriched int eh ribosomal fraction, and by how much. Ago1? Any other RNAi components?

8. L182-184 – I found something about the wording of this sentence to be opaque. I had to read it several times.

9. Could the authors confirm that neither virus tested encodes a VASR that is functional in Drosophila, and could affect the experiment or its interpretation?

10. Line 234 – What is a canonical tube?

11. L271 ‘douncer’ is rather informal, if cute. The ‘Dounce homogeniser’ is named for Alexander Dounce

PLOS authors have the option to publish the peer review history of their article (what does this mean? ). If published, this will include your full peer review and any attached files.

**Do you want your identity to be public for this peer review?** For information about this choice, including consent withdrawal, please see our Privacy Policy .

Reviewer #1: No

Reviewer #2: No
---

## [Editor Report · Decision Letter 1]

17 Dec 2024

Dear Dr. Marques,

We are pleased to inform you that your manuscript 'Argonaute 2 targets viral transcripts but not genomes of RNA viruses during antiviral RNA interference in Drosophila' has been provisionally accepted for publication in PLOS Pathogens.

Best regards,

Benhur Lee

Section Editor

PLOS Pathogens

Benhur Lee

Section Editor

PLOS Pathogens

Sumita Bhaduri-McIntosh

Editor-in-Chief

PLOS Pathogens

orcid.org/0000-0003-2946-9497

Michael Malim

Editor-in-Chief

PLOS Pathogens

orcid.org/0000-0002-7699-2064
---

## [Editor Report · Acceptance letter]

Dear Dr. Marques,

We are delighted to inform you that your manuscript, "Argonaute 2 targets viral transcripts but not genomes of RNA viruses during antiviral RNA interference in Drosophila," has been formally accepted for publication in PLOS Pathogens.

Best regards,

Sumita Bhaduri-McIntosh

Editor-in-Chief

PLOS Pathogens

orcid.org/0000-0003-2946-9497

Michael Malim

Editor-in-Chief

PLOS Pathogens

orcid.org/0000-0002-7699-2064